# Quantifying the Performance of Conflict-free Replicated Data Types in InterPlanetary File System

Quentin Acher
Université de Lorraine, CNRS, Inria, LORIA
Nancy, France
quentin.acher@inria.fr

Claudia-Lavinia Ignat
Université de Lorraine, CNRS, Inria, LORIA
Nancy, France
claudia.ignat@inria.fr

Shadi Ibrahim
Inria, Univ Rennes, CNRS, IRISA
Rennes, France
shadi.ibrahim@inria.fr

## Abstract

The continuous growth in data volume increases the interest in using peer-to-peer (P2P) systems not only to store static data (i.e., immutable data) but also to store and share mutable data – data that are updated and modified by multiple users. Unfortunately, current P2P systems are mainly optimized to manage immutable data. Thus, each modification creates a new copy of the file, which leads to a high "useless" network usage. Conflict-free Replicated Data Types (CRDTs) are specific data types built in a way that mutable data can be managed without the need for consensus-based concurrency control. A few studies have demonstrated the potential benefits of integrating CRDTs in the InterPlanetary File System (IPFS), an open-source widely used P2P content sharing system. However, they have not been implemented and evaluated in a real IPFS deployment. This paper tries to fill the gap between theory and practice and provides a quantitative measurement of the performance of CRDTs in IPFS. Accordingly, we introduce IM-CRDT, an implementation of CRDTs in IPFS that focuses on the simple data type (i.e., Set); and carry out extensive experiments to verify whether CRDTs can efficiently be utilized in IPFS to handle mutable data. Experiments on Grid'5000 show that IM-CRDT reduces the data transfer of an update by up to 99.96% and the convergence time by 54.6%-62.6%. More importantly, we find that IM-CRDT can sustain low convergence time under concurrent updates.

*CCS Concepts:* • **Information systems → Distributed storage**; **Data replication tools**; • **Networks → Network measurement**.

*Keywords:* P2P Storage Systems, Replication, Consistency, CRDT, IPFS, Performance Evaluation

**ACM Reference Format:**
Quentin Acher, Claudia-Lavinia Ignat, and Shadi Ibrahim. 2023. Quantifying the Performance of Conflict-free Replicated Data Types in InterPlanetary File System. In *Proceedings of DICG 2023*. ACM, New York, NY, USA, 6 pages. https://doi.org/10.1145/3631310.3633488

## 1 Introduction

In current cloud storage services, data are replicated to ensure high availability in case of failures. Specifically, multiple copies of the same data are stored in a few machines normally placed in different geographical locations, so if a failure occurs data are still accessible in other replicas. A key question is how to maintain consistency between all replicas [2, 10]. Most of the current cloud storage systems use a central server to synchronize replicas. However, this solution does not scale up with the number of replicas and their frequency of modification. To eliminate the disadvantages of a central entity that has to perform the synchronisation and to ensure easy deployment, and resilience to failures and attacks, alternative solutions, using Peer-to-Peer (P2P) technologies, were proposed [4].

Meant to be "the storage layer of the decentralized web", the InterPlanetary File System (IPFS) [9] is an open-source P2P content addressed file system. It allows people to share data, either directly or via programs that use IPFS as a library. In IPFS, data are immutable by design and modifying an object requires creating a new modified object. IPFS does not offer support for merging concurrent changes, i.e., users are not able to update concurrently the replicas of the same data without losing their modifications.

As stated by the CAP theorem [2], high availability, low latency and data consistency are difficult to achieve in distributed systems in the presence of network partitions. In order to ensure high data availability, consistency is relaxed. Eventual consistency [3, 10] is a weak consistency model where replicas are allowed to diverge and they will converge later after the reception of all updates. One of the main family of replication algorithms for ensuring convergence under eventual consistency is Conflict-free Replicated Data Types (CRDTs) [6]. The main idea is to define data structures where parallel modifications are conflict free. CRDTs respect Strong Eventual Consistency, a property that ensures convergence as soon as every replica has integrated the same modifications without further message exchange among replicas.

Several studies have discussed the feasibility of using CRDTs in P2P systems, including IPFS. LogootSplit [7] is a sequence CRDT that was applied for building a P2P web based real-time collaborative editor [11]. In [12], the authors formalised Merkle-CRDTs and discussed the benefits of integrating them in IPFS. While previous work evaluated CRDTs in the context of collaborative editing [5], no implementation and evaluation of CRDTs was done in a real IPFS deployment. To this end, in an attempt to fill the gap between theory and practice, this paper provides – to the best of our knowledge – the first quantitative measurement of the performance of CRDTs in IPFS. Specifically, we introduce IM-CRDT, an implementation of CRDTs in IPFS that focuses on the simple data type (i.e., Set). Experiments on Grid'5000 [8] show that IM-CRDT reduces the data transfer of an update by up to 99.96% and the convergence time by 54.6%-62.6%. More importantly, we find that IM-CRDT can sustain low convergence time under concurrent updates. To summarize, we make the following contributions.

*DICG 2023, December 2023, Bologna*
© 2023 Association for Computing Machinery.
ACM ISBN 978-x-xxxx-xxxx-x/YY/MM...$15.00
https://doi.org/10.1145/3631310.3633488

- We present an implementation of CRDTs in IPFS named IM-CRDT, which targets specifically the Set data type.
- We conduct extensive experiments to verify whether CRDTs can efficiently be used in IPFS to handle mutable data. The results clearly demonstrate the effectiveness of IM-CRDT in maintaining low convergence time under concurrent updates.
- We open-sourced IM-CRDT and made all the scripts and experimental results available at: https://github.com/Mutable-Data-Over-Peer-to-Peer-Systems/IM-CRDT.

The paper is structured as follows. Section 2 presents a brief overview of IPFS and CRDTs. Section 3 describes the integration of CRDTs into IPFS. The performance evaluation of this integration is presented in section 4. Section 5 concludes the paper and outlines future work.

## 2 Background

This section first briefly introduces IPFS, and then presents Merkle-clocks and CRDTs.

**IPFS.** IPFS [9] is a P2P version-controlled file system that enables data sharing among users. IPFS's protocols define its behavior, such as how to address data through Content IDentifier (CID), how to share and store data and how to manage security issues and replication. A file is presented as a Merkle Directed Acyclic Graph (Merkle-DAG), where each node refers to a shard of the data. A user informs other users that it shares a specific file by publishing a Provider Record on the Distributed Hash Table (DHT) that contains the link between the file CID and the Peer Identifier. The naming protocol allows users to share a file that can be modified only by its creator, hence other peers can only read it. IPFS is efficient and widely used [14]. However, it is not meant to deal with mutable data that can be concurrently modified, as each modification requires creating a whole new file.

**Merkle-clocks.** In distributed systems, logical clocks are generally used for ordering events [1]. A logical clock is a structure $c$ that associates to every event $e$ a value $C(e)$ that creates a partial order respecting the causal order. Merkle-clocks [12] are Merkle-DAG-based logical clocks where nodes represent events. The order relation between events is defined by the existence of a path between nodes.

**CRDTs.** CRDT is a data structure that ensures Strong Eventual Consistency without explicit coordination. There are two main families of CRDTs: state-based and operation-based [6]. They differ in the way payloads are defined, i.e., how the updates are shared. A payload under state-based CRDT contains the whole data, while the payload under operation-based CRDT carries only a single update. Merkle-CRDTs [12] are based on Merkle-clocks where each node of the Merkle-DAG refers to CRDT payloads.

## 3 Integration of CRDTs in IPFS

In this section, we explain how we implemented the data replication mechanism and the communication mechanism among the replicas. We used the *Go* implementation of IPFS, named kubo[1], to implement IPFS as a file sharing system, and used libP2P's communication protocols. In our evaluation, we focus on the Set data type. However,

---

[1] https://github.com/ipfs/kubo

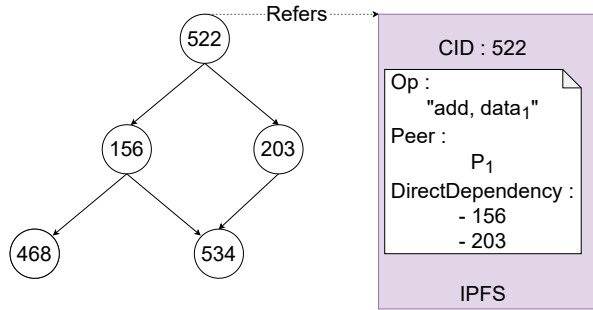

**Figure 1.** Example of a CRDT DAG

it is important to note that the modules for the definition of Merkle-CRDT nodes and for Merkle-CRDT merging can be easily extended to other CRDTs.

### 3.1 Merkle-CRDT's Nodes

Nodes of the Merkle-CRDT's DAG are built to be easily converted into a file which will can be then shared through IPFS. This is done by encoding the node's structure into a JSON file. The file must contain all the necessary information needed by a peer after downloading it. A node contains all the CIDs of its direct sons in the DAG, we refer to them as the *direct dependencies*. The file also contains the Peer IDentifier (PID) of the peer that emitted the event, and the payload of the event it carries, i.e., the data shared by the event in a CRDT format. Figure 1 shows an operation-based CRDT DAG with data that was modified five times, once per node. The root node *522* refers to a file stored in IPFS containing the following information: the operation of the update, i.e., an *add* operation , followed by the data added; the peer identifier of the publisher, $P_1$; the *Direct dependencies* which include the root nodes that were known by the publisher $P_1$ when publishing the *add* operation.

### 3.2 Merkle-CRDT's DAG

The Merkle DAG is defined by a set of root nodes (that do not have parents). The data state can be obtained by means of a recursive traversal of the DAG, starting from these root nodes. Note that when concurrent updates occur, we can have multiple root nodes. Indeed, multiple updates can be concurrently done and will therefore have the same direct dependency. For instance, in Figure 1, before the creation of the node *522*, the Merkle-CRDT's DAG had two root nodes: *156* and *203*. Once a node is read by IPFS, it is stored locally in a file. Hence, each node in Merkle-CRDT refers to a local file containing the updated data.

### 3.3 Merkle-CRDT's communication

The communication module among peers was implemented by means of a publish-subscribe mechanism. A publish-subscribe topic is created by a specific peer named the bootstrap peer. The bootstrap peer opens a socket where any peer can connect in order to join the publish-subscribe topic and subscribe to it. When a peer applies an update to the data, it creates a file corresponding to the update's node. Then the file is added to IPFS and the peer registers the corresponding CID that will be sent to all other peers through the publish-subscribe topic. On the receiver side, when a peer receives a CID from the topic, it saves the CID in a file, so later an

active loop can read it. If the CID is unknown, it will search for the corresponding data in IPFS, and write it in a new local file. When the corresponding data is received, two cases must be handled. If all the *direct dependencies* are already known, the node is directly added to the Merkle-DAG. Otherwise, the system will recursively download all the dependencies. Such download might introduce an overhead when the Merkle-DAG becomes large, as confirmed in [12].

Figure 2 represents an example of an update managed by our system. The system applies updates through three remote steps, and a fourth local step. At the beginning, *Peer 1* prepares the node's information in a file including the performed operation, the peer creating it, and the CID of the direct dependencies. In step 1, the updater *Peer 1* sends these data to IPFS. IPFS computes the data's CID and announces it to the other peers. In step 2, libP2P's pubsub protocol is used to send a message containing the node's CID to the other peers. In step 3, each peer that receives the CID retrieves the data from an IPFS provider, stores it in a new local file, and locally maps this CID to this new file. Two cases must be handled. If *direct dependencies* are unknown, they need to be retrieved from the IPFS network. When retrieving node 156, *Peer 3* knows all dependencies, while *Peer 2* needs to retrieve node 534. In step 4, each peer adds locally the discovered node to its Merkle-DAG. Finally, we can notice that *Peer 2*' Merkle-DAG has two root nodes 794 and 156, the node 794 representing a concurrent update which has not yet been sent. When this concurrent update is sent and integrated by the other peers, the Merkle-DAGs of all peers will converge. Note that the order of updates' reception and therefore the order of local files creation storing these updates are different from one peer to another. For instance, "F4.json" represents different CIDs in *Peer* 2 and *Peer* 3 corresponding to different updates.

The system described in this section shares mutable files over IPFS by using Merkle-CRDTs and manages concurrency.

## 4 Evaluation

In this section, we evaluate the performance of integrating simple string Set CRDTs in IPFS. Our evaluation demonstrates that:

- Compared to the default IPFS implementation, IM-CRDT improves convergence time and reduces network usage. IM-CRDT reduces the data transfer of an update by up to 99.96% and the convergence time by 54.6%-62.6%.
- IM-CRDT can sustain low convergence time under concurrent updates. However, sequential downloading becomes a bottleneck when the update rate is high.
- IM-CRDT does not introduce high overhead (compute time) when integrating files into Merkle-CRDTs, under concurrent updates.

### 4.1 Performance metric

We quantify the feasibility and effectiveness of our system based on the convergence time, i.e., the time period between an update is issued and when it is integrated in the Merkle-CRDT DAGs of all replicas. We denote the time period for an update $x$ issued by a peer $p_j$ to be successfully received by a peer $p_i$ as $\underset{p_j \rightarrow p_i}{update\ Latency(x)}$. It is composed of the time required by the publisher to issue the corresponding file in IPFS ($t_{add}^x$), the time for transmitting the CID from the publisher to the receiver ($t_{pubsub}^x$), the time to retrieve the

IPFS file and unknown dependencies ($t_{retrieve}^x$), and the time to integrate the file in the receiver's local Merkle-CRDT DAG ($t_{compute}^x$). $\underset{p_j \rightarrow p_i}{update\ Latency(x)}$ can be computed as:

$$
\begin{aligned}
\underset{p_j \rightarrow p_i}{update\ Latency(x)} &= t_{add}^x + t_{pubsub}^x \\
&\quad + t_{retrieve}^x + t_{compute}^x \\
&= t_{p_i}^x - t_{p_j}^x
\end{aligned}
$$

where $t_p^x$ is the time at which the peer $p$ creates/receives $x$.

We define the convergence time for an update $x$ issued by $p_{i_0}$ as:

$$
\begin{aligned}
maximum\ latency(x) &= \max_{1 \leq i \leq n;\ i \neq i_0} \left( t_{p_i}^x - t_{p_{i_0}}^x \right) \\
&= \max_{1 \leq i \leq n;\ i \neq i_0} \left( \underset{p_j \rightarrow p_i}{update\ Latency(x)} \right)
\end{aligned}
$$

where $(p_i)_{\{0 \leq i \leq n-1\} \setminus \{i_0\}}$ are the peers hosting the replica.

### 4.2 Compared solutions.

We evaluated two approaches: the default implementation of IPFS, and IM-CRDT ( InterPlanetary Merkle-CRDT).

- **IPFS** is used as a baseline approach when updates are issued sequentially. IPFS sends the whole file to all other peers after each update. We do not evaluate IPFS under concurrent updates because this requires to implement a locking mechanism in IPFS which may introduce significant overhead and scalability limitations.
- **IM-CRDT** is our implementation which integrates CRDTs with IPFS. We implemented string Set CRDTs based on the grow-Only Set structure [6], supporting string addition (*Add*) and removal (*Remove*). When adding a string $x$ in the set $S$ ($S = (S_{Add}, S_{Remove})$), the addition operation is applied $S_{Add} = S_{Add} \cup \{x\}$; and when removing a string, the removal operation is applied $S_{Remove} = S_{Remove} \cup \{x\}$. Grow-Only Sets maintain *Strong Eventual Consistency*, ensuring replicas convergence regardless of the order of execution of $(ADD, x)$ and $(REMOVE, x)$ operations. We use operation-based CRDTs, as they can easily manage incremental modifications. A unique identifier is associated to every element added into the Set.

### 4.3 Experimental setup

**Experimental testbed.** Our experiments were conducted on the French scientific testbed Grid'5000 [8] at the site of Nantes. We use the *econome* and *ecotype* clusters with 66 machines. Each machine in *econome* is equipped with 2 Intel Xeon E5-2660 8-core processors, 64 GB of main memory, and one HDD at 7.2k RPM with 2 TB. The machines in *ecotype* are equipped with 2 Intel Xeon E5-2630L 10-core processors, 128 GB of main memory, and one SSD with 400 GB. The machines are connected by 10 Gbps Ethernet network. The two TORs switches of both clusters are connected with four 40 Gbps links. All machines run 64-bit Debian stretch Linux with GO 1.20.4, IPFS kubo v0.19.0, libP2P v0.26.4 and libP2P-pubsub v0.9.3. All the experiments were done in isolation on the testbed, with no interference originated from other users. The results are the mean of 3 runs.

**Deployment.** In each experiment, we start by initiating $n$ nodes/peers with $n \in \{2, 5, 10, 20, 50\}$ nodes/peers. Their clocks are then synchronized using the NTP protocol. We select one peer to act as a bootstrap peer. We wait for 1 minute (this time is set to 5 minutes

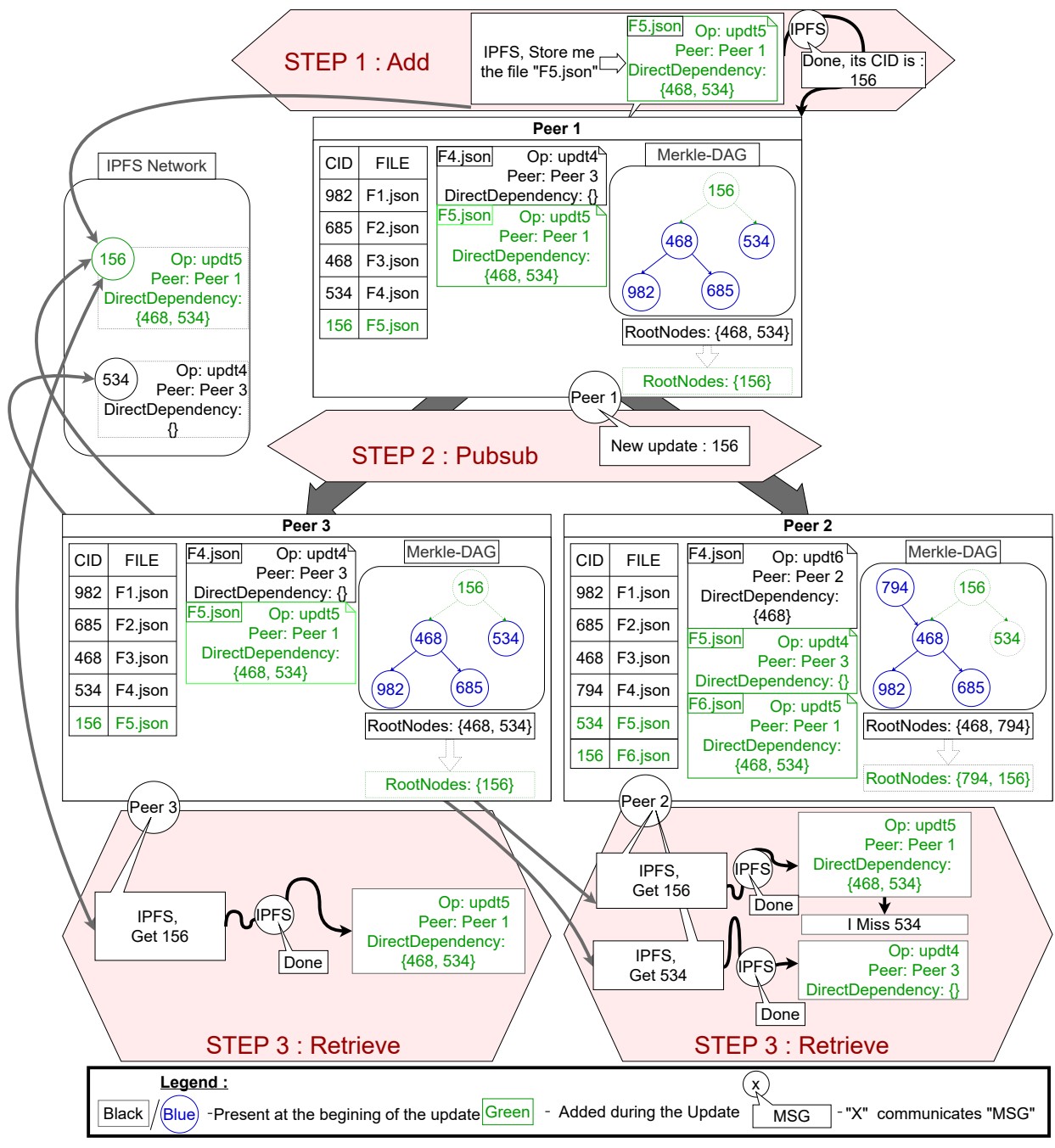

**Figure 2.** Updates under IM-CRDT

in the case of 5 replicas) – to make sure that all peers are connected – and then we start sending out the updates. In the case of sequential updates, we configured the bootstrap peer to send the updates. And in the case of concurrent updates, the bootstrap peer and $n_{peerUpdating} \in \{1, 4, 9, 19\}$ peers are configured to send updates. We save the $t^x_{add}$ in the updating nodes; and the $t^x_{retrieve}$ and $t^x_{compute}$ in the receiving nodes. For the $t^x_{pubsub}$, we compute it using the following equation:

$$t^x_{pubsub} = \underset{p_j \rightarrow p_i}{update\ Latency(x)} - t^x_{add}$$
$$- t^x_{retrieve} - t^x_{compute}$$

**Workload.** We start the experiment with a 1 $MB$ file. Then, each updater sends one update per second. In each update, a unique string is added. This string is a growing integer signed by the date and the peer identifier.

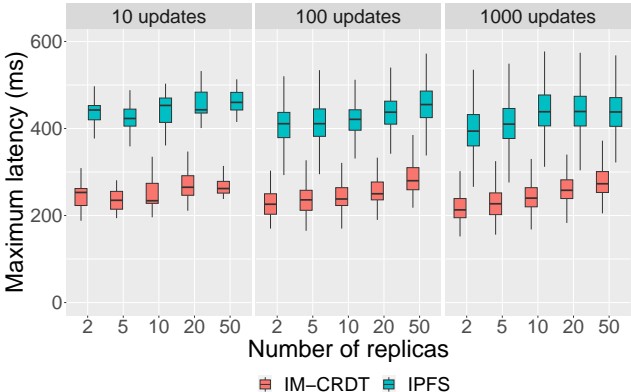

**Figure 3.** Maximum latency for each update using *IM-CRDT* and *IPFS*

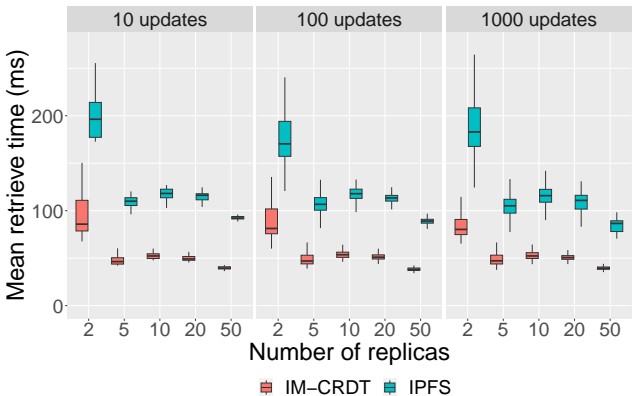

**Figure 4.** Mean retrieve time for an update using *IM-CRDT* and *IPFS*

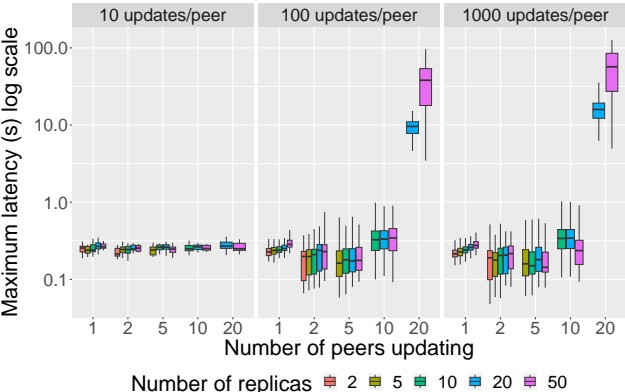

**Figure 5.** Maximum latency for concurrent updates using *IM-CRDT*

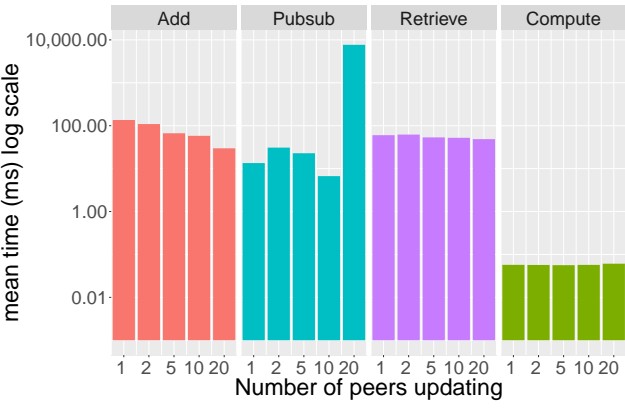

**Figure 6.** Breakdown of the different steps for the updates in *IM-CRDT* with 20 Replicas

### 4.4 Results

**IM-CRDT *vs.* IPFS.** Figure 3 shows how IM-CRDT compares with IPFS with respect to convergence time (maximum latency). We vary the number of replicas from 2 to 50, and the number of updates from 10 to 1000. We can see that IM-CRDT can reduce the maximum latency by 54.6%-62.6% compared to IPFS, on average. The reason behind that is the reduction in the retrieval time under IM-CRDT compared to IPFS, as shown in Figure 4, which is due to the significant reduction in the size of updates. The average size of an update is only 350 bytes with *IM-CRDT*, while it is 1 MB with *IPFS* – IPFS sends the whole file to all replicas after each update.

**The impact of concurrency in IM-CRDT.** Figure 5 shows the changes in maximum latency of IM-CRDT under concurrent updates. Interestingly, for most cases, IM-CRDT can sustain low maximum latency when increasing the number of concurrent updates. For example, the latency increases by only 6%, from 249 ms to 264 ms when increasing the number of concurrent updates from 1 to 20, with 10 updates. However, we noticed that the maximum latency increases significantly, by 133% and 205%, when the number of concurrent updates is set to 20 and the total number of updates is 100 and 1000, respectively. Taking a closer look at the results, Figure 6 shows the breakdown of the different steps for the updates, with

20 replicas. We observe that pubsub time is considerably higher when having 20 concurrent updates. As a reminder, the pubsub time ($t^x_{pubsub}$) is the time from when the updater starts sending the update's CID to the time when the receivers start downloading the update. This increase in pubsub time is not caused by libP2P's pubsub mechanism, but it is the result of the sequential downloading of updates in IPFS. An update takes about $50 - 100$ ms so when 20 updates arrive at the same time, it takes more than one second for the retrieve step to be completed, during which the next batch of updates should wait.

**Overhead of IM-CRDT: Compute time.** Figure 7 shows the compute time when varying the number of concurrent updates. IM-CRDT introduces a slight increase in the compute time (by up to 7%), reaching 60.8 $\mu$s for 20 updates.

## 5 Conclusion and Future Work

This paper presents IM-CRDT, an implementation of CRDTs in IPFS for the Set data type. Our evaluation results show that IM-CRDT can reduce the amount of data transfer and achieve fast convergence time. In future work, we plan to evaluate our implementation using more complex replicated data types including sequences [7] and relational databases [13]. We also plan to investigate how to handle

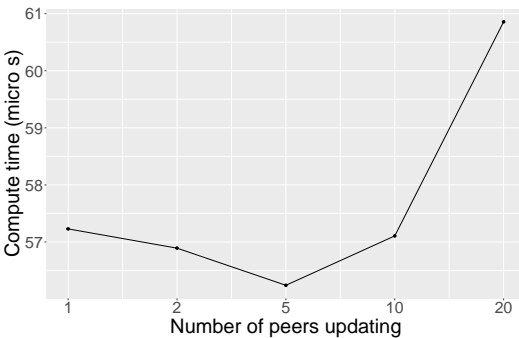

**Figure 7.** Evolution of compute time when the number of concurrent updates grows

efficiently concurrent downloads under a high rate of concurrent updates.

## Acknowledgments

This work is supported by the "Alvearium" Inria and hive partnership. Experiments presented in this paper were carried out using the Grid'5000 testbed, supported by a scientific interest group hosted by Inria and including CNRS, RENATER, and several Universities as well as other organizations (see http://www.grid5000.fr/).

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
