# OpenReview forum: "Quantifying the Performance of Conflict-free Replicated Data Types in InterPlanetary File System"
_ACM.org/Middleware/Workshop/DICG — DICG 2023_

### Official Review · Reviewer_PCVV · 2023-10-22
**Quantifying the Performance of Conflict-free Replicated Data Type in Interplanetary File System**

**Rating:** 6
**Confidence:** 5

**Review:**

Presentation of the paper is well below standard and should in and onto itself qualify the paper for rejection. The authors must carefully proof-read and if they are not able to determine writing issues themselves, either use a native speaker or use online tools to help with corrections (e.g., grammarly).

Now having said that, there also are some noteworthy points about the work and paper.

The paper describes a novel method to enact incremental changes for files in IPFS. Since files in IPFS are immutable, modifying files is achieved by creating new versions of the same file, thus incurring a waste of resources. With IM-CRDT, data are modeled by a collection of files as op-based CRDT updates. Furthermore, these files are organized into a Merkle-CRDT DAG to ensure all modifications are captured. IM-CRDT increased the performance of modifying files in IPFS by reducing repetitive traffic and enabling peers to apply concurrent updates.

Three or more strong points about the paper:
1.	 The CRDT solution to the stated problem is simple yet effective, as shown by the evaluations.
2.	Good number of evaluations for a short paper.
3.	The workflow of the system is well explained.

Three or more weak points about the paper:
1.	There are many typos, grammar mistakes, and missing punctuation.
2.	The logic flow of the paper is confusing.
3.	There is no discussion if there are other attempts to solve the stated problem.
4.	The current work only includes Set CRDT.

The paper proposes a novel solution using existing techniques (CRDTs) in an existing system (IPFS). The current solution is limited (set CRDT), but it can potentially be more widely applicable.

Detailed comments to the authors:
-	Many writing mistakes (below are ones that I observed):

o	Section 1: Ment to be "the storage layer of the decentralized web"

o	In IPFS, data is immutable by design and modifying an object requires creating a new modified object.

o	users are not able to update concurrently the replicas of the same data without loosing their modifications.

o	Section 2: Each CRDT family defines differently the payload,

o	Section 3: We used the Go implementation of IPFS named kubo1 implementing (to implement) IPFS as a file sharing system.

o	Section 4 (Evaluation) is not in the past tense.

-	Many incoherent sentences

-	The paper lacks an overview or intuition of the proposed system. In Section 3, the first paragraph talks about implementation details, then moves directly to components of the system (Merkle-CRDT’s Nodes, DAG, etc). It is very hard to grasp, even the concept itself is straightforward.

-	The overall logical flow can be improved.

-	There is no discussion of whether there are other attempts to solve the problem of updating files in IPFS, as this seems to be a significant problem. If there are, they should be compared against the IM-CRDT.

-	How to make it more widely applicable? The paper only demonstrates set CRDT, but IPFS is a generic file system that supports any type of file.

-	The system can be seen as using IPFS to distribute CRDT updates; what are the benefits compared to just using a reliable P2P protocol?

-	What is the consistency guarantees of IM-CRDT?

-	Since updates to a CRDT are persistent through the IPFS, one additional feature that can be added is the undoing of previous updates by checking the content and applying an inverse update. See “Mao, Yunhao, Zongxin Liu, and Hans-Arno Jacobsen. "Reversible conflict-free replicated data types." Proceedings of the 23rd ACM/IFIP International Middleware Conference. 2022.”

---

### Official Review · Reviewer_yiWx · 2023-10-30
**This paper implements Conflict-free Replicated Data (CRDT) in the Interplanetary File System targeting the Set data type. The authors conducted experiments to gather information and measure the performance of the solution.**

**Rating:** 5
**Confidence:** 3

**Review:**

## Pros

- IPFS stores immutable data. CRDTs would allow storing mutable data, a feature desirable for a general-purpose storage system.
- The paper brings insight into the practical aspects of a potential IPFS-CRDT integration with a concrete implementation (IM-CRDT) and a deployment of various experiments to measure the latency of individual and concurrent updates, mean retrieve time, and computation time.
- The IM-CRDT significantly reduces the data transfer of an update since the vanilla version of IPFS requires sending the whole file every time the file is updated.

## Cons

- I don't find sufficient novelty in this paper. Merkle-clocks were presented previously in the cited paper [9].
- Introducing CRDT creates important inefficiencies when downloading all the updates of a file. This limitation is already discussed in [9].  The authors perform measurements that illustrate the limitation but do not bring any solution to the problem.
- The positive results regarding the reduction in the retrieval time due to the small size of the updates are not surprising.

## Others
- Figure 2 has errors. Peer 2 includes 794 as child and root. F4.json in peer 2 is different from the JSON file that other peers downloaded. It is unclear "who is speaking" inside the dialogue globes.
- Typo: "ment to be"